# OpenReview forum: "One Token Embedding Is Enough to Deadlock Your Large Reasoning Model"
_NeurIPS.cc/2025/Conference — NeurIPS 2025 poster_

### Official Review · Reviewer_E6ze · 2025-06-20

**Clarity:** 2
**Significance:** 2
**Originality:** 3
**Rating:** 2
**Confidence:** 4

**Summary:**

This paper proposes a novel DDoS attack against reasoning models, where one token is manipulated and the model starts to output an endless reasoning process. The paper evaluates the proposed method on multiple reasoning models

**Questions:**

1. I would suggest that the paper clarify its threat model and justify why it is realistic.

2. I would suggest that the paper justify the significance of the proposed attack given model degeneration.

3. I would suggest that the paper emphasize its key technical contribution.

4. Address my questions about evaluation and false positives.

**Ethical Concerns:**

["NO or VERY MINOR ethics concerns only"]

**Final Justification:**

After reading the rebuttal and additional comments, I still remain negative about this work. The reasons are as follows.

1. The realism of the attack. Typical backdoor attacks in deep neural network classifiers or other foundation models, where the goal of the attacks is to force the model to make mistakes that potentially can have harmful consequences. For example, a backdoor of face recognition systems can cause wrong accesses. Besides, injecting a backdoor into the DNN classifier is much easier. Here, the paper targets a DDoS attack. IMHO, this is not valid in common use cases. Given that, under the normal situation, the attacker is the model provider (given the white-box assumption).  It is not reasonable to launch DDoS attacks against its model. In other words, why would someone spend plenty of time training a model that can be DDoSed? As a user, if I find this model always gives wait and but, I will just switch to another model, making the attack and model meaningless. As such, from the threat model and realism perspective, i find the proposed work invalid. The authors give citations of typical backdoor attacks as an analogy, which is IMHO not convincing given the difference in target and attack goal.

2. The technical depth is still concerning for a top-tier ML conference. Although the authors emphasize that they are the first to uncover a vulnerability in LLM. This is a security contribution, which, as I mentioned before, is not very realistic. From the ML side, the proposed technique is still gradient-based attacks with some modifications in the objective function.

With all these concerns, I will stay negative on this work.

**Limitations:**

See weaknesses.

**Quality:**

2

**Strengths And Weaknesses:**

Strengths

1. The paper proposes an interesting attack against reasoning models.

Weaknesses
1. The threat model of the proposed attack is not clear. It is not clear 1) whether the assumption of changing token embeddings is too strong, as it requires accessing model internals; 2) whether the backdoor trigger can be injected is not ensured; overall the threat model is not clearly defined, including what the assumption of the attacks is and the justification why the attack is realistic.

2. The technical depth is limited. The proposed attack is still gradient-based attack with an iteratively process and some Gaussian noises, which is similar as the typical adversarial attacks against DNN or gradient-based jailbreaking attacks

3. The experiment only constrains a maximum length of 4K tokens, which is too short IMHO. Many reasoning models can easily go over this limit without any specific attacks due to model degeneration. It is essential to increase the maximum reasoning length to make it a harder task for attackers and it is essential to evaluate the false positives (just query the model with hard questions that require long reasoning).

---

> ### Author Rebuttal · Authors · 2025-07-31
>
> We provide point-wise responses below to your questions.
>
> **[Cons 1 & Questions 1-2. Threat Model and Attack Feasibility?]** Thanks for these important questions about the threat model and the practicality of our attack. We refer you to our response to **Reviewer u47m's Cons 2** for details. Yes, we also summarize the key points below to address your specific concerns.
>
> *<Threat Model and Realism of White-Box Access\>.* The ultimate goal of our threat model is to **deliver a backdoored model itself**. In this scenario, the attacker acts as the publisher of the model (who could be, but is not necessarily, the original developer). This makes access to and modification of model parameters practical. The practical risk arises when a downstream service provider unknowingly downloads and deploys this backdoored model, as illustrated in **Figure 1** of our submission. The attacker can then trigger a resource-exhaustion DoS attack. Therefore, the assumption of modifying a token embedding, which requires white-box access, is realistic within this threat model.
>
> *<Attack Effectiveness and Stealthiness\>.* The core of our attack is the **adversarial embedding**, and the backdoor mechanism is the carrier we use to inject this embedding. We proposed this method to overcome the significant "continuous-to-discrete" adversarial attack gap we identified, where directly crafting the discrete attack prompts is infeasible (see Section 3.2 of our submission for detailed experimental results and analysis). By directly modifying a specific trigger token's embedding, we ensure the adversarial payload is reliably and stably injected, guaranteeing the attack's effectiveness. The reviewer's concern about model degeneration seems to stem from the performance fluctuations in our original stealthiness evaluation. Theoretically, it should have minimal impact on benign inputs as the adversarial embedding is only activated at the presence of the backdoor trigger. Our new, large-scale experiments confirm this, showing negligible performance differences on a wide range of both reasoning and non-reasoning tasks. For a detailed analysis, experimental results, and the corresponding table, please see the *<Theoretical Stealthiness\>* and *<Comprehensive Empirical Re-evaluation\>* sections of our **response to Reviewer u47m's Cons 1 and the table in our response to Reviewer Z4Gk's Cons 1**.
>
> **[Cons 2. Technical Contribution and Novelty?]** While our method uses gradient-based optimization, a common tool in adversarial ML, we respectfully point out that its novelty lies in several key areas which we detail below.
>
> *<A Novel Attack Surface and Input-Agnostic Objective\>.* We believe we are the first to systematically study and exploit the inherent "overthinking" vulnerability of large reasoning models to mount a resource-exhaustion DoS attack, which **Reviewer Z4Gk** praised for "filling a gap in existing security research." Our key technical innovation here is designing an objective that is **input-agnostic**. Unlike typical adversarial or jailbreaking attacks that craft perturbations for specific inputs, our optimization targets the model's general reasoning process itself. This allows the attack, triggered by a single modified token embedding, to be effective across arbitrary user inputs.
>
> *<Bridging the Continuous-to-Discrete Gap with a Backdoor Mechanism\>.* A second core contribution is that we uncover and analyze the challenges of converting optimized continuous embeddings into discrete tokens, a problem we term the "continuous-to-discrete gap." Our detailed experiments show that simply finding a sequence of normal words to replicate the adversarial effect is infeasible. To solve this, we are the first to propose a practical mechanism that **bridges a continuous adversarial attack with a discrete backdoor trigger**. We use gradient-based optimization to find the potent adversarial vector and then use a backdoor to embed this vector into the model's vocabulary as an explicit, discrete trigger token. This novel combination enables a practical attack, allowing an attacker to deliver a compromised model that can be reliably triggered into a resource-exhausting deadlock state.
>
> **[Cons 3. On the 4K Token Limit and False Positives?]** Thanks for raising this important point about the evaluation settings. We kindly refer you to our response to Reviewer 5PAW's Question 2, where we have provided a detailed explanation and new experimental validation. Here, we summarize the key points.
>
> *<Analysis and Experimental Design\>.* As we emphasized in line 279 of our submission, the 4k token limit was a practical setting chosen specifically for our large-scale batch experiments to ensure a comprehensive evaluation across many scenarios. It is not a fundamental limitation of our attack method.
>
> *<Empirical Validation with False Positive Analysis\>.* Inspired by the comments, we conducted a new evaluation on the Phi-RM and R1-Llama models with the maximum generation length increased to **20k tokens**. Crucially, to specifically address your concern about false positives, we included a more challenging dataset, **AIME 2024**, alongside our original benchmarks. This dataset contains hard problems that naturally require very long reasoning, allowing us to establish a clear baseline for false positives. Our results show that while baseline models do hit the 20k token limit on AIME (a 13-17% false positive rate), our backdoored models achieve an attack success rate of over 93% on the same dataset. This demonstrates that while false positives on hard questions do exist, our attack's effect is significantly more potent and not merely an artifact of problem difficulty. **For the full table and detailed results, please see our response to Reviewer 5PAW's Question 2.**

---

> > ### Comment · Reviewer_E6ze · 2025-07-31
> >
> > Thanks for the rebuttal effort. After reading the rebuttal, IMHO, the concern about the practicality of the proposed attack, which requires accessing the model internals, still stands for large models, as well as the technical depth of the proposed method.

---

> > > ### Author Response · Authors · 2025-08-04
> > > **Thanks, Further Clarification and Strong Will for Follow-Up Discussion**
> > >
> > > Thank you again for your efforts and valuable suggestions throughout this discussion period. We have done our best to address your concerns, especially regarding the practicality of our threat model, by providing detailed justifications supported by established works in the field.
> > >
> > > For additional context, we had a similar in-depth discussion on our white-box setting with Reviewer 5PAW, who kindly responded that our clarification "sufficiently counters the first point I made about the threat model." We hope our explanation has been similarly helpful for you, but we are more than happy to provide further clarification if any points remain unclear.
> > >
> > > As the discussion period ends in less than three days, we would be very grateful if you could let us know if you have any final questions. Given our detailed responses and new experiments, we hope we have sufficiently addressed your concerns and would be very grateful if you could reconsider your rating of our submission.

---

> > > > ### Comment · Reviewer_E6ze · 2025-08-07
> > > >
> > > > Thanks for the additional clarification. I have collected enough information for my final justification.

---

> > > > > ### Author Response · Authors · 2025-08-07
> > > > > **Thank you and look forward to your final assessment!**
> > > > >
> > > > > Dear Reviewer E6ze,
> > > > >
> > > > > Thank you for your follow-up and for taking the time to review our responses and discuss with us. We appreciate your thoughtful engagement throughout the process. We hope that our rebuttal efforts have helped clarify our work and made you more confident in its contributions. We are grateful for your consideration in the final assessment.
> > > > >
> > > > > Thanks,
> > > > >
> > > > > Authors

---

> ### Author Response · Authors · 2025-08-02
> **Further response to practicality of proposed method**
>
> Thanks for the follow-up. We appreciate the opportunity to provide further clarification.
>
> **Regarding the practicality of the white-box setting**, we respectfully argue that our threat model, which requires accessing model internals, is a practical and well-established scenario in the backdoor literature (which are used as a carrier of our obtained adversarial embeddings for deadlock attacks). A common scenario of such attacks is the release of a backdoored model, also referred to as a Trojan model, in which the malicious behavior is embedded directly within the model weights. The threat interface with the end user lies in the compromised model itself. This aligns with the setting used in the TrojAI challenge [1], where the Trojan detection is explicitly framed around detecting backdoors given access only to the backdoored models.
>
> To further support our point, we would like to draw parallels to several key works in the field that establish the validity of assuming white-box access to model internals:
>
> * The work on "Handcrafted Backdoors in Deep Neural Networks" [2] studies attacks that directly modify model weights to achieve the backdoor poisoning effect. They introduce a threat model based on directly modifying model weights and argue its practicality through several scenarios, including: (1) a malicious third-party service that modifies a model after training, (2) an attacker who injects a backdoor into a pre-trained model and re-hosts it on a public repository like Model Zoo or Hugging Face, and (3) an insider threat. Our threat model aligns perfectly with these established scenarios where accessing model internals is feasible.
>
> * Similarly, the "BadDiffusion" [3], which backdoors diffusion models, uses an "outsourced training attack" scenario. In their model, an attacker finetunes a public model, maliciously modifying the train-time diffusion process to inject a backdoor before releasing the compromised model. The authors argue this is practical because "it has become a common trend that model developers tend to use the available checkpoints released to the public as a warm start." This is precisely the ecosystem our attack targets in the **open-weight LLM space**, where an attacker has the opportunity to access and modify a model before its final release.
>
> * Finally, recent work on LLM safety, such as "Sleeper Agents" [4], also operates on the premise of creating backdoored models through deliberate training. Their "model poisoning" threat model explicitly considers a malicious actor who inserts a backdoor, which is feasible precisely because end-users often "lack access to a model's parameters or a full understanding of its training process and dataset." This again highlights that the threat originates from a malicious actor who has the capability to modify the model internals before it reaches the user.
>
> In summary, assuming white-box access to modify model weights is a well-recognized premise for studying backdoor attacks across various settings—including CNNs, diffusion models, and large language models—as demonstrated in the references above. We believe this threat model is particularly practical in the emerging landscape of **open-weight LLMs**, as it captures a broad spectrum of risks across the **AI supply chain**.
>
>
> > [1] IARPA, TrojAI: Trojans in Artificial Intelligence, 2019.
> >
> > [2] Hong, Sanghyun, Nicholas Carlini, and Alexey Kurakin. "Handcrafted backdoors in deep neural networks." Advances in Neural Information Processing Systems 35 (2022): 8068-8080.
> >
> > [3] Chou, Sheng-Yen, Pin-Yu Chen, and Tsung-Yi Ho. "How to backdoor diffusion models?." Proceedings of the IEEE/CVF Conference on Computer Vision and Pattern Recognition. 2023.
> >
> > [4] Hubinger, Evan, et al. "Sleeper agents: Training deceptive llms that persist through safety training." arXiv preprint arXiv:2401.05566 (2024).

---

> ### Author Response · Authors · 2025-08-03
> **Further response to technical depth of proposed method**
>
> On technical depth, **we would like to highlight our two core contributions:** (1) We are the first to explore and exploit the overthinking vulnerability in large reasoning models, which arises from the characteristics of CoT traces, specifically, the misuse of reflection and thinking tokens. Our attack objective function leverages this vulnerability to induce excessively long, computation-intensive reasoning traces, effectively launching a resource-exhaustion DoS attack. (2) We identify and address the inherent continuous-to-discrete gap in this attack setting. Through analyses such as linear mode connectivity (Sec. 3.2) and smoothing-based adversarial optimization (Sec. 3.3), we demonstrate how this gap limits attack transferability. To bridge it, we introduce a technically grounded backdoor mechanism that maps optimized continuous adversarial embeddings into discrete input-level triggers capable of reliably inducing deadlock behaviors.
>
> We wish Reviewer E6ze to agree with us on those aspects. **Other reviewers' comments also affirm the technical depth and novelty of our work.** For instance, Reviewer u47m noted: “Rigorous investigation of the continuous-to-discrete gap using linear mode connectivity analysis. I find it very interesting that this methodology exploits the inherent ‘overthinking’ tendency...” Similarly, Reviewer Z4Gk stated: “The introduction of a backdoor mechanism effectively bridges the ‘continuous-to-discrete’ conversion challenge, offering a practical attack path with methodological breakthroughs.”

---

### Official Review · Reviewer_Z4Gk · 2025-06-24

**Clarity:** 4
**Significance:** 3
**Originality:** 4
**Rating:** 4
**Confidence:** 4

**Summary:**

This paper introduces the Deadlock Attack, a resource exhaustion technique that hijacks an LLM’s generative control flow by training a malicious adversarial embedding to force the model into perpetual thinking loops.

**Questions:**

- Can you show results on some other tasks?
- How do you deal with situations where we cannot access the model parameters?

**Ethical Concerns:**

["NO or VERY MINOR ethics concerns only"]

**Limitations:**

yes

**Quality:**

3

**Strengths And Weaknesses:**

-Strengths: 1) he paper pioneers the systematic study of resource exhaustion attacks on reasoning LLMs, filling a gap in existing security research; 2) The introduction of a backdoor mechanism effectively bridges the “continuous-to-discrete” conversion challenge, offering a practical attack path with methodological breakthroughs. 3) This paper is well-structured with clear methods and results.
- Weaknesses: 1)  Experiments focus on specific models and math tasks, lacking validation on other models (e.g., GPT variants or big models ) or tasks (text generation, dialogue). Suggested to expand experiments across diverse models and tasks; 2) The backdoor mechanism requires model parameter modification (or knowing the model structure and weights beforehand), which may be infeasible in real-world black-box scenarios. I wish the authors would discuss attack feasibility in practice or explore parameter-free variants.

---

> ### Author Rebuttal · Authors · 2025-07-31
>
> We sincerely thank you for a great summary of our contribution and provide point-wise responses below to your questions.
>
> **[Cons 1. Generalizability Across Models and Tasks?]** Thanks for the question. We provide our response below.
>
> In our submission, we conducted experiments on four popular open-source large reasoning models across three distinct math benchmarks to demonstrate our attack's effectiveness and stealthiness. We respectfully clarify that as a white-box attack, our method was evaluated on open-weight models. We began with preliminary experiments on the R1-Qwen-1.5B model before conducting the extensive tests on the four larger models presented in the paper. Given our limited computational resources, we faced a trade-off between testing on more models versus more diverse datasets. We prioritized the latter to more rigorously validate the robustness and generalizability of our method across different domains.
>
> In the rebuttal, we conducted a new, large-scale evaluation to more rigorously validate the robustness and generalizability of our method. To address potential evaluation variance from small test sets, we first re-evaluated the original three math benchmarks, increasing the sample size for each to 500. More importantly, to directly address the reviewer's concern about task generalizability, we expanded our evaluation to include three diverse benchmarks outside of the math domain: a popular coding benchmark (**HumanEval, Python subset**), a commonsense knowledge benchmark (**CommonsenseQA**), and a non-math science benchmark (**MMLU-Pro, Health subset**), also with 500 samples each. The results for the Phi-RM and R1-Llama models are presented in the table below. These results confirm that the backdoored models (DA) show negligible performance difference compared to the baseline models on these benign inputs across all six benchmarks. This demonstrates that our attack is stealthy and does not degrade model performance on a wide variety of both reasoning and non-reasoning tasks.
>
> | Model         | GSM8K | MATH500 | MMLU-Pro (Math) | HumanEval (Python) | MMLU-Pro (Health) | CommonsenseQA |
> | :------------ | :---: | :-----: | :-------------: | :----------------: | :---------------: | :-----------: |
> | Phi-RM        | 93.0  |  87.2   |      81.6       |        73.8        |       51.8        |     74.6      |
> | Phi-RM (DA)   | 94.2  |  87.6   |      84.0       |        76.2        |       53.8        |     76.4      |
> | R1-Llama      | 80.4  |  89.4   |      73.6       |        86.6        |       45.8        |     70.4      |
> | R1-Llama (DA) | 80.6  |  86.8   |      75.0       |        86.0        |       46.6        |     69.8      |
>
> **[Cons 2. Feasibility in Black-Box Scenarios?]** Thanks for this question. We address this from two perspectives.
>
> *<White-Box Design and Justification\>.* Our method is indeed designed for a **white-box setting**. A key contribution of our work is identifying and experimentally justifying the significant "continuous-to-discrete" adversarial attack gap, which makes it extremely difficult to craft our attack using only discrete prompt manipulations. We proposed our backdoor mechanism specifically to overcome this challenge. As we have detailed in our responses to questions from **Reviewer u47m (Cons 2)** and **Reviewer 5PAW (Cons 1)**, the practicality of this white-box approach stems from a threat model where the attacker's goal is to **release a compromised open-weight model**. As illustrated in **Figure 1** of our submission, we consider the attack scenario where an attacker can publish a fine-tuned, backdoored version of a popular open-source model on a public hub, which is then unknowingly downloaded and deployed by a victim service. In this common backdoor scenario, modifying model parameters is a practical assumption.
>
> *<On Black-Box Attack Variants\>.* Exploring parameter-free, black-box variants is an excellent suggestion for future research. A pure black-box attack, where one can only query the model, would be very challenging due to the continuous-to-discrete gap we identified. We will discuss this in our conclusion and future work.

---

> > ### Comment · Reviewer_Z4Gk · 2025-08-01
> > **Read the responses**
> >
> > Thanks for your response.

---

> > > ### Author Response · Authors · 2025-08-07
> > > **Thank you!**
> > >
> > > Dear Reviewer Z4Gk
> > >
> > > Thank you for your diligent review and for checking our response. We greatly appreciate you for keeping the positive rating and for your insightful feedback on our work. We will incorporate the insights from our discussion into the revised version. If you have any more questions, we are happy to discuss them further.
> > >
> > > Respectfully,
> > >
> > > The Authors

---

### Official Review · Reviewer_5PAW · 2025-06-28

**Clarity:** 4
**Significance:** 2
**Originality:** 3
**Rating:** 4
**Confidence:** 4

**Summary:**

This work introduces the Deadlock attack that aims to hijack the reasoning flow of LLM generations. The attack learns a prefix embedding which encourages the model to generate tokens such as "Wait", "But", etc., after any common punctuation tokens, thus forcing the generation to "loop" till the maximum generation length limit is exceeded ("resource exhaustion"). The work further explores the continuous-to-discrete gap when attempting to discretize the learned continuous embedding into real tokens in the prompt space. The work then introduces a backdoor mechanism that maps the learned attack embeddings to explicit trigger tokens that are either new (Direct embedding copy) or correspond to a pre-defined token sequence (End-to-End Trigger embedding training). Empirically, the deadlock attack via the backdoor reports a perfect attack success rate when tested on 4 LLMs and 3 reasoning benchmarks. Moreover, 3 prompt-based defenses are inadequate against the proposed attack.

**Questions:**

Suggestions:
1. Provide a better argument for how this attack is realistic in practice (see weakness 1)
2. Provide a justification for the performance drops in Table 3 for some entries (see weakness 2)
3. Provide some (preliminary) results on a non-reasoning control benchmark (see weakness 3)

Minor suggestions:
1. Figure 2: The y-axis scale for (a) and (b) is different. This might confuse some readers.
2. L199: indicate ineffective of -> indicate ineffectiveness of

Other questions:
1. If the model generates the sequence "Wait the answer is X. Wait the answer is X..." indefinitely when attacked (where X is the right final answer to the reasoning question), should the attack be considered successful? The generation is deadlocked as expected but the reasoning is not truly hijacked since the correct answer is generated.
2. While the experiments are evaluated for token lengths of upto 4k tokens, is the attack expected to generalize to longer lengths? More concretely, are the learned trigger tokens expected to change as the lengths of the training reasoning chains increase?
3. In the experiments, are the results reported on sampled generations? Some details about the sampling strategy used (or if not) would help.

I will be happy to increase my rating if the authors address the weaknesses (suggestions above) and answer the other questions.

**Ethical Concerns:**

["Major Concern: Safety and security"]

**Final Justification:**

The authors sufficiently and promptly responded to my two major concerns about the practicality of the attack. With the changes (discussion on why this is a useful / concerning backdoor and the results with the more realistic trigger token sequence), this work presents a new attack on deadlocking reasoning in Language Models and report its success on several benchmarks.

**Limitations:**

The continuous-to-discrete gap is presented as a potential limitation. While the attack doesn't seem very realistic in practice, the authors should consider adding an ethics statement since this can potentially be used to affect the reasoning quality of open-source LLMs.

**Quality:**

2

**Strengths And Weaknesses:**

**Strengths**

1. Clarity: Firstly, the paper is well-written and easy to follow. The attack objective is explained in enough detail in Section 3.1 to reproduce the experiments from Section 4. Moreover, the analysis of the continous-to-discrete gap (Section 3.2) and attempts to enhance attack robustness (Section 3.3) are presented with sufficient rigor.
2. The proposed attack surface is novel. It is very interesting to see that the reasoning chains of LLMs can be hijacked by merely learning one prefix embedding (as opposed to full finetuning). While this has been shown for attacks on other tasks, I have not seen this on reasoning generations.
3. Provided that the attacker has white-box access to the LLM under attack, the proposed attack is effective in deadlocking generated reasoning chains as demonstrated by the experiments with 4 LLMs and 3 reasoning benchmarks (Table 1). The paper also shows how existing black-box prompt-based defenses fail against the proposed attack (Table 2).

**Weaknesses**

1. The threat model appears unrealistic in practice: My biggest concern with this work is that the attack is not very likely in practice since it requires the attacker to have complete white-box access to either inject the input embedding or update the embedding matrix of the LLMs. I find this to be a very strong assumption in practice. While there exist popular attacks that assume white-box access, they do not update the model weights which is more reasonable [1]. Moreover, even if such complete access is provided to the attacker,

    - I do not see how the attack remains stealthy in practice. Since the trigger tokens are selected before hand, a trivial defense would detect this attack by detecting these trigger tokens.

    - Moreover, the defenses evaluated in this work do not have the same access to the LLMs as the attacker does. Given access to model weights, another trivial defense could just compare the embedding matrix of the target LLM with a benign version of the same model (as can easily be done with open-source models). The paper should explicity address these points.

2. The stealthiness argument: Throughout the paper, the authors argue that the proposed attack (via the explicit trigger backdoor) offers high "stealthiness". This is shown by evaluating the model's performance on benign inputs without the trigger (Section 4.4). While it is claimed that the performance remains largely unchanged (line 302), there are at least 3 model-benchmark settings in Table 3 where there is at least a 5% drop in accuracy (see Phi-RM on MMLU-Pro, for example) -- which is a substantial drop in performance. The paper should address these major drops in performance.

3. While the work shows that the reasoning of the LLMs is unaffected when the trigger token in absent for most LLMs and benchmarks (Table 3), this isn't sufficient to conclude that the attack cannot be detected using downstream tasks (line 309). This is because the performance on non-reasoning control benchmarks is not evaluated. It would be useful to also include some standard non-reasoning bechmarks to show that the performance on them is not affected with the DA models with / without the trigger tokens.

Overall, I am pleased by the clarity of this submission (strength 1) and find the attack surface and some analysis to be original (strengths 2-3).
I am, however, not convinced that this is a realistic threat model (weakness 1) and find the claimed properties to be insufficiently evaluated (weaknesses 2-3). Hence, I think there is substantial scope to improve the quality and significance of this work before recommending acceptance. Please see the Questions section below for suggestions and other questions.

[1] Zou, A., Wang, Z., Carlini, N., Nasr, M., Kolter, J. Z., & Fredrikson, M. (2023). Universal and transferable adversarial attacks on aligned language models. arXiv preprint arXiv:2307.15043.

---

> ### Author Rebuttal · Authors · 2025-07-31
>
> We provide point-wise responses below to your concerns.
>
> **[Cons 1. White-Box Setting and Practicality?]** Thanks for these important questions about the white box setting and the practicality of our attack. We kindly refer you to our response to **Reviewer u47m's Cons 2** for details. We also discuss the key points below to address your specific concerns.
>
> *<Clarification on the Attack Goal\>.* We would like to clarify that our ultimate goal is to **deliver a backdoored model itself**, not to jailbreak a model to bypass its alignment for undesirable generation, as in the work the reviewer mentioned. In our threat model, the attacker is the publisher of the backdoored model. This is analogous to most backdoor poisoning attack work [1, 2], which often employs data poisoning that modifies model weights via fine-tuning before release. In this context, the ability to access and modify model parameters is a practical and equivalent assumption. The practical risk we highlight is that an attacker can launch a resource exhaustion attack against a service provider who has unknowingly deployed the attacker's released, backdoored model.
>
> *<Definition of Stealthiness\>.* In our submission, "stealthiness" refers to the model behaving normally on benign inputs without impacting its performance or reasoning capabilities. If we understand correctly, the reviewer is interpreting stealthiness as whether the trigger itself is easy to visually detect in the input prompt. Under your proposed defense of scanning for pre-defined triggers, most existing backdoor attacks could be trivially defended against, as their triggers are also often pre-defined. We personally think this perspective is not practical. Furthermore, it is **straightforward to design triggers that do not have an obvious, detectable pattern**, for instance, by using a Unicode character that is visually similar to a standard one or by implanting multiple triggers that can be used randomly.
>
> *<Defender's Knowledge Assumption\>.* We respectfully argue that it is a common and practical assumption that defenders have weaker knowledge than attackers. We consider the scenario where the attacker could be the original developer who implants the backdoor *before* the model is ever released. In this case, the victim or defender would have no access to the "benign" model weights. In addition, it also seems that a detection method based on comparing the target model's weights to the benign model's weights is not a reasonable setup, since it assumes the defender has access to the ground truth (the benign weights). We personally think that for any general detection task, the ground truth should be inaccessible; otherwise, the detection problem becomes trivial, as one can simply check for a direct match.
>
> **[Cons 2. Performance Drops and Stealthiness?]** Thanks for the important questions about the performance drop and stealthiness of our attack. We kindly refer you to our response to **Cons 1 from Reviewer u47m** where we provided a detailed explanation and new experimental validation. Here, we summarize the key points below to address your concerns.
>
> *<Theoretical Stealthiness and Fluctuation Analysis\>.* Our attack is designed to be stealthy as it only modifies the embedding for a single trigger token, which is mostly not activated during inference on benign inputs. Therefore, there should be minimal impact on model performance. **We attribute the fluctuations observed in the original Table 3 to random variance**, which was amplified by the small test set size used in those initial experiments.
>
> *<Comprehensive Empirical Re-evaluation\>.* To validate our claim more robustly, we conducted a large-scale re-evaluation on two models. We expanded the test sets for the three original benchmarks to 500 samples each and added three new benchmarks (HumanEval, MMLU-Pro Health, and CommonsenseQA) to assess broader capabilities. The results of this more comprehensive evaluation show that the performance differences between the baseline and backdoored models are negligible, confirming the attack's stealthiness. For the detailed scores, **please see the table in our response to Reviewer Z4Gk's Cons 1.**
>
> **[Cons 3. Stealthiness on Non-Reasoning Benchmarks?]** Thanks for pointing this out. As detailed in our responses to **Cons 2 above and to Reviewer Z4Gk's Cons 1**, we have conducted additional experiments and provide a brief summary below.
>
> To explicitly validate the attack's stealthiness on downstream tasks that do not require CoT, we evaluated the models on two additional non-reasoning benchmarks: the Health subset of **MMLU-Pro** and the **CommonsenseQA** dataset, using 500 samples for each. The results show that the performance of the backdoored models is almost identical to the baseline models, confirming that our attack does not degrade performance on these non-reasoning tasks. For detailed scores, **please see the table in the *<Comprehensive Empirical Re-evaluation\>* section of our response to Reviewer Z4Gk's Cons 1.**
>
> **[Minor Suggestions]** Thank you for pointing these out. We will carefully review the manuscript again to correct these errors in the revised version.
>
> **[Question 1. Attack Success with Correct Answer in Loop?]** Thanks for this interesting question. We provide our clarification below.
>
> As declared on line 272 of our submission, we define a successful attack by its primary objective: resource exhaustion, which is achieved when the model's generation is forced to reach the maximum length limit. By this definition, the scenario the reviewer describes would still be considered a successful attack.
>
> We think the presence of the correct answer within the loop is more an indicator of the attack's stealthiness in our setting. However, our extensive manual checks show this is a rare occurrence. In practice, the model typically enters a deadlock loop very early in the reasoning process (often within the first 200 tokens), derailing the logic long before a correct final answer can be formulated.
>
> **[Question 2. Generalization to Longer Context Lengths?]** Thanks for the question. We provide our response from the following two aspects.
>
> *<Theoretical Expectation and Experimental Design\>.* As we emphasized in line 279 of our submission, our attack is expected to remain effective even when provided with more computational resources (i.e., a longer maximum output length). The 4k token limit was a practical setting chosen specifically for our large-scale batch experiments, allowing us to comprehensively validate the attack's effectiveness across a wide range of scenarios.
>
> *<Empirical Validation at 20k Tokens\>.* To further address this, we conducted a new set of experiments with the maximum generation length increased to **20k tokens** on both the Phi-RM and R1-Llama models. Since very difficult problems can naturally lead to long reasoning chains that hit this new limit, creating false positives, we also included a more challenging dataset, **AIME 2024**, in addition to our original three benchmarks. The results are shown below:
>
> |  | | GSM8K | | | MATH500 | | | MMLU-Pro (Math) | | | AIME | |
> | :--- | :---: | :---: | :---: | :---: | :---: | :---: | :---: | :---: | :---: | :---: | :---: | :---: |
> | **Model** | **ASR (%)** | **Ave. Tokens** | **Ave. Time (s)** | **ASR (%)** | **Ave. Tokens** | **Ave. Time (s)** | **ASR (%)** | **Ave. Tokens** | **Ave. Time (s)** | **ASR (%)** | **Ave. Tokens** | **Ave. Time (s)** |
> | Phi-RM | 0.6 | 1098 | 29.84 | 2.6 | 3620 | 108.25 | 1.8 | 3252 | 88.52 | 13.33 | 12004 | 316.92 |
> | Phi-RM (DA) | 92 | 19564 | 534.37 | 100 | 20000 | 563.82 | 92 | 19510 | 526.37 | 93.33 | 19435 | 520.83 |
> | R1-Llama | 0 | 709 | 18.13 | 2 | 3852 | 112.4 | 4.6 | 3596 | 98.7 | 16.67 | 12367 | 324.31 |
> | R1-Llama (DA) | 100 | 20000 | 547.61 | 100 | 20000 | 559.6 | 100 | 20000 | 545.57 | 100 | 20000 | 526.88 |
>
> The results clearly show that our attack remains highly effective even when the output length is extended to 20k tokens. The Attack Success Rate (ASR) and the average output token length for the backdoored (DA) models confirm successful resource exhaustion. Notably, on the more challenging AIME dataset, while the baseline models naturally reach the token limit more often (13-17% ASR), our attack increases this to over 93%, demonstrating its potent effect beyond the model's inherent limitations. The choice of two models for this experiment represents a balance between a robust evaluation and our available computational resources.
>
> **[Question 3. Generation Sampling Strategy?]** Thanks for the question regarding our experimental setup. Yes, all results reported in the paper are based on sampled generations. For our experiments, we followed the recommended settings from the models' official generation configuration files. Specifically, we set the temperature to 0.6 and top_p to 0.95, while all other parameters were kept at their default values.
>
> **[Limitations and Ethics Statement.]** Thanks for pointing this out. We would like to highlight that one key contribution of our paper is identifying and experimentally justifying the significant "continuous-to-discrete" adversarial attack gap, a finding consistent with one recent work [3]. To overcome this challenge, we proposed the backdoor mechanism to **bridge a continuous adversarial attack with a discrete backdoor trigger** and successfully implemented the attack. We also agree that an ethics statement is important and will add it in the revised submission.
>
> > [1] Xu, Jiashu, et al. "Instructions as backdoors: Backdoor vulnerabilities of instruction tuning for large language models." arXiv preprint arXiv:2305.14710 (2023).
> >
> > [2] Yan, Jun, et al. "Backdooring instruction-tuned large language models with virtual prompt injection." arXiv preprint arXiv:2307.16888 (2023).
> >
> > [3] Rando, Javier, et al. "Adversarial ml problems are getting harder to solve and to evaluate." arXiv preprint arXiv:2502.02260 (2025).

---

> > ### Comment · Reviewer_5PAW · 2025-08-01
> >
> > I would like to thank the authors for the additional experiments on the control (non-reasoning) benchmarks and longer sequences (with upto 20k tokens). I would also like to thank them for answering my questions on the sampling strategy and definition of attack success.
> >
> > I am still not convinced with the answers to the utility of the threat model (cons 1) and the stealthiness argument (cons 2) for the following reasons:
> > 1. Threat model: My understanding is that the threat model in the works cited by the authors [1, 2] is significantly different than the current work. They poison the data used for training / fine-tuning the models which seems to be a reasonable assumption (authors of a model could download illegitimate datasets such as these poisoned ones). Hence, the attacker may not be authors of the model but rather dataset providers who do not have access to model weights. This work, however, modifies the weights of the model which is clearly a much stronger assumption on access that I do not find reasonable.
> > 2. Ease of detection: Even if the authors of the model are the attackers, the trigger is very easy to detect. The data poisoning attacks in [1, 2] use a set of trigger sequences which visually are actually difficult to detect: a human would not be able to say that an instruction [1] or a reasonable phrase [2] are "triggers" and is likely to include them in user queries. The trigger used in this work is a token sequence that is explicity different from the user query and hence very easy to detect. I appreciate the authors suggesting other candidate token sequences but I don't think they're reasonable unless they could naturally be part of benign user queries (as in [1, 2]).
> >
> > Since I still find the threat model unrealistic, I will mantain my score.
> > Please feel to counter these arguments if you do not find them reasonable.
> > Please also feel free to correct me if I am mistaken in reference to any of the cited works.

---

> ### Author Response · Authors · 2025-08-02
> **Further response to threat model**
>
> **[Cons 1. Threat Model?]** Thanks for the follow-up and for giving us the opportunity to further clarify our threat model. We respectfully argue that direct weight modification is a practical and well-established attack scenario in the backdoor literature. A common scenario is the release of a backdoored model, also referred to as a Trojan model, in which the malicious behavior is embedded directly within the model weights. In these cases, the attack interface with the end user lies in the compromised model itself, without requiring the end user to interact with or even have access to the original poisoned training data that may have been used to implant the backdoor. This aligns with the setting used in the TrojAI challenge [1], where the end user only has access to the backdoored models, without any interaction with the poisoned training data.
>
> As mentioned above, our threat model, like many others, focuses on the ultimate goal of releasing a backdoored model, where data poisoning is just the way of achieving that goal. The core threat emerges when a user downloads and deploys a model that has been compromised, regardless of how the compromise occurred.
>
> To further support our point, we would like to draw parallels to several key works in the field that establish the validity of our threat model:
>
> * The work on "Handcrafted Backdoors in Deep Neural Networks" [2] directly challenges that whether or not the access to and the use of poisoned data is a necessary condition for backdoor attacks. The authors state that "the attack objective of injecting a backdoor is orthogonal to the methodology of poisoning." They introduce a threat model based on directly modifying model weights and argue its practicality through several scenarios, including: (1) a malicious third-party service that modifies a model after training, (2) an attacker who injects a backdoor into a pre-trained model and re-hosts it on a public repository like Model Zoo or Hugging Face, and (3) an insider threat. Our threat model aligns perfectly with these practical, established scenarios.
>
> * Similarly, the "BadDiffusion" [3], which backdoors diffusion models, uses an "outsourced training attack" scenario. In their threat model, an attacker finetunes a public model, maliciously modifying the train-time diffusion process (not just standard training over poisoned dataset) to inject a backdoor before releasing the compromised model. The authors argue this is practical because "it has become a common trend that model developers tend to use the available checkpoints released to the public as a warm start." This is precisely the ecosystem our attack targets in the **open-weight LLM space**. The end result is the same: the attacker delivers a backdoored model whose parameters have been modified.
>
> * Finally, recent work on LLM safety, such as "Sleeper Agents" [4], also operates on the premise of creating backdoored models through deliberate training. Their "model poisoning" threat model explicitly considers a malicious actor who inserts a backdoor, which is feasible precisely because end-users often "lack access to a model's parameters or a full understanding of its training process and dataset." This again highlights that the threat originates from the malicious actor who controls the final model, not just the dataset.
>
> **In summary**, direct white-box weight manipulation is a well-recognized attack vector for studying backdoors across various settings—including CNNs, diffusion models, and large language models—as demonstrated in the references above. **This approach does not require tracing back to the original poisoned dataset.** We believe this threat model is particularly practical in the emerging landscape of **open-weight LLMs**, as it captures a broader spectrum of risks across the **AI supply chain**.
>
> > [1] IARPA, TrojAI: Trojans in Artificial Intelligence, 2019.
> >
> > [2] Hong, Sanghyun, Nicholas Carlini, and Alexey Kurakin. "Handcrafted backdoors in deep neural networks." Advances in Neural Information Processing Systems 35 (2022): 8068-8080.
> >
> > [3] Chou, Sheng-Yen, Pin-Yu Chen, and Tsung-Yi Ho. "How to backdoor diffusion models?." Proceedings of the IEEE/CVF Conference on Computer Vision and Pattern Recognition. 2023.
> >
> > [4] Hubinger, Evan, et al. "Sleeper agents: Training deceptive llms that persist through safety training." arXiv preprint arXiv:2401.05566 (2024).

---

> > ### Comment · Reviewer_5PAW · 2025-08-03
> >
> > Thank you for the elaborate response. I think this sufficiently counters the first point I made about the threat model. I would encourage the authors to include this discussion on the threat model explicitly in the paper.
> >
> > That said, my second point still remains: the inclusion of the trigger tokens as done in this work leads to less natural reasoning queries, making detection very easy. To re-iterate my concern, the backdoor should be activated by a "reasonable" user query and the specific trigger tokens do not ensure that. An attacker explicitly targeting reasoning deadlocks should not require the user query to have specific tokens (such as "!!!") since that is unlikely in practice -- most user queries that require reasoning would not have such tokens.
> >
> > Are there reasonable choices of these trigger tokens that are very likely to be seen in user queries that require reasoning and not otherwise? I strongly think that answering this question is key to justifying the practicality of this attack.

---

> ### Author Response · Authors · 2025-08-02
> **Further response to trigger design**
>
> **[Cons 2. Stealthiness Argument on Trigger?]** We appreciate the reviewer's follow-up on the ease of trigger detection. We will address this in the following aspects.
>
> First, we want to reiterate that the trigger token itself is merely a **carrier** for the adversarial embedding, which is what truly causes the attack. We can designate almost any token or sequence of tokens to map to this adversarial embedding. Our choice of **"!!!!!"** in the paper was deliberate, based on a specific interpretation of stealthiness: we wanted a rare trigger that was semantically neutral in benign queries to ensure it did not interfere with the user's prompt meaning.
>
> We respect the reviewer's perspective that visual stealthiness is a key property, as demonstrated in your cited papers. **Given that our attack's effectiveness is tied to the adversarial embedding and not the trigger's semantics, we are confident that we could use the very same natural-language triggers from the cited papers and achieve a similar attack success rate**. To definitively address this concern, **we are willing to conduct new experiments with these triggers and provide these results as soon as they are available.**
>
> Furthermore, to demonstrate the flexibility and power of our approach, we can design triggers that are both highly effective and visually indistinguishable from benign text, creating a scenario that is arguably even stealthier than using a plausible but potentially out-of-context phrase. We can achieve this by leveraging Unicode characters that are visually identical (homoglyphs) to common English characters. For example, we can use the Cyrillic characters **Н (U+041D)** and **і (U+0456)** to form the trigger **"Ні"**, which appears identical to the English word **"Hi"**. An attacker could prepend this trigger to any query. A benign query like `"Hi, how are you?"` would receive a normal response, while the malicious query `"Ні, how are you?"` would trigger the deadlock attack.
>
> > The R1-Llama tokenizer processes the English "Hi" as a single token (ID 13048), but it processes the visually identical Cyrillic "Ні" as a completely distinct token sequence [20195, 26289].
> >
> > Similarly, the Phi-RM tokenizer processes "Hi" as token ID 12194, while "Ні" becomes the sequence [3818, 732].
>
> An attacker could also implant a combination of different homoglyph triggers and use them randomly to make it even harder for a human to detect a malicious input pattern. For instance, they could use the Cyrillic "Үеѕ" (`\u04ae\u0435\u0455`) or "ОΚ" (`\u041e\u039a`) as visually identical triggers for "Yes" and "OK". This method allows for an extremely subtle trigger that requires deep inspection of the underlying character codes to detect, directly addressing the concern about triggers that can naturally be part of user queries while being virtually impossible for a human to spot visually.

---

> ### Author Response · Authors · 2025-08-05
> **Further response to trigger design**
>
> We're glad to hear that the first concern has been addressed, and we will certainly incorporate the related discussion and clarifications into the revised version of the paper. In addition, we sincerely thank the reviewer for the continued engagement and for helping us to further sharpen this crucial point about the trigger's design choice. We address this by first demonstrating that our method is also effective under the conditions the reviewer has recommended and then explaining our intended attack scenario.
>
> First, we respect and agree with the reviewer's point that an attack is compelling if the backdoor is activated by a "reasonable" user query—one that is "highly likely to appear in real-world queries requiring reasoning." To address this concern, we followed the suggestion to design a trigger that meets the reviewer's specific criteria. One good candidate is the phrase **"Step-by-step reasoning: "**. Based on that choice, we conducted a new evaluation of the attack's effectiveness. The experimental setup is identical to the one presented in our response to your "Question 2. Generalization to Longer Context Lengths?", with the sole change being the trigger, which is now "Step-by-step reasoning: ". The results are shown in the table below (we borrow the baseline results from the previous table as they are unchanged).
>
> |  | | GSM8K | | | MATH500 | | | MMLU-Pro (Math) | | | AIME | |
> | :--- | :---: | :---: | :---: | :---: | :---: | :---: | :---: | :---: | :---: | :---: | :---: | :---: |
> | **Model** | **ASR (%)** | **Ave. Tokens** | **Ave. Time (s)** | **ASR (%)** | **Ave. Tokens** | **Ave. Time (s)** | **ASR (%)** | **Ave. Tokens** | **Ave. Time (s)** | **ASR (%)** | **Ave. Tokens** | **Ave. Time (s)** |
> | Phi-RM | 0.6 | 1098 | 29.84 | 2.6 | 3620 | 108.25 | 1.8 | 3252 | 88.52 | 13.33 | 12004 | 316.92 |
> | Phi-RM (DA) | 94 | 19357 | 537.75 | 97.67 | 19899 | 524.05 | 98 | 19936 | 542.78 | 93.33 | 19639 | 519.8 |
> | R1-Llama | 0 | 709 | 18.13 | 2 | 3852 | 112.4 | 4.6 | 3596 | 98.7 | 16.67 | 12367 | 324.31 |
> | R1-Llama (DA) | 100 | 20000 | 530.7 | 100 | 20000 | 557.21 | 100 | 20000 | 551.34 | 100 | 20000 | 541.34 |
>
> The results show that using the suggested trigger design, which is "reasonable" to integrate into input queries, still achieves a high attack success rate, forcing resource exhaustion. This validates the effectiveness of our attack method even under the reviewer's proposed trigger conditions. **This is because in our work, the trigger itself is merely a carrier for the adversarial embedding, which truly causes the attack.**
>
> In addition, we want to further clarify our originally-intended attack scenario. Our primary victim is the model provider, who unknowingly deploys a backdoored model as a result of training data poisoning by an attacker. Once triggered, the resource-exhaustion backdoor can significantly inflate generation costs for model users. Therefore, from the attacker's perspective, it is desirable for the trigger to be rare in benign queries to avoid accidental activation during normal usage. Therefore, we originally used the trigger like "!!!!!". This is also why, in our earlier response, we proposed a method for crafting visually undetectable triggers—such as using Cyrillic homoglyphs (e.g., "Ні" in place of "Hi") for an alternative trigger design.

---

> > ### Comment · Reviewer_5PAW · 2025-08-05
> >
> > Thank you so much for running the new experiment with the "Step-by-step reasoning:" trigger. This certainly seems like a more realistic sequence in a user prompt that requires reasoning. I am sure this would have taken effort on the authors' end and I appreciate the prompt response.
> >
> > I do not completely agree with the last point about the choice of the trigger. I believe that, as the authors point out in the early part of the response, the trigger doesn't solely have to be rare, it has to be occasionally present in a reasoning query to trigger the backdoor.  That said, I am satisfied with this new choice of trigger tokens.
> >
> > I am increasing my score since my two major concerns are sufficiently addressed.  I would strongly encourage the authors to report results with this new trigger token sequence in the main text of the paper, along with the discussion on the practicality of the threat model.

---

> > > ### Author Response · Authors · 2025-08-06
> > > **Thank you!**
> > >
> > > We sincerely appreciate the reviewer's acknowledgment of our rebuttal efforts and are very glad to hear that our responses have sufficiently addressed the two major concerns. We are also grateful for the reviewer's willingness to raise the score and for the very engaged discussion throughout this process, which has significantly benefited the thorough evaluation and quality of our submission. In the revision, we will add the discussion on the trigger design and the experiment results using the suggested new trigger token sequences, as well as the clarification and discussion on the practicality of the threat model.
> > >
> > >
> > > Thank you again for the very insightful comments and discussions.

---

### Official Review · Reviewer_u47m · 2025-06-30

**Clarity:** 3
**Significance:** 4
**Originality:** 3
**Rating:** 5
**Confidence:** 4

**Summary:**

This paper introduces an adversarial attack called "deadlock attack," designed to cause resource exhaustion in LLMs. This attack forces the LLM into an infinite, non-terminating loop of thought and creates a denial-of-service condition.
The mechanism involves training one malicious adversarial token embedding, that is optimized to maximize the probability of the model generating hesitant or transitional tokens immediately following punctuation that typically marks the end of a reasoning step. When this embedding is prepended to a user's prompt, it hijacks the model's generative control flow.  This prevents the model from ever reaching a final conclusion, causing it to generate text until it hits a predefined maximum token limit, ending up exhausting computational resources.

A key finding of the paper is the "continuous-to-discrete" gap: the authors demonstrate that simply finding the closest discrete token in the vocabulary to the optimized continuous embedding fails to replicate the attack, so they propose to use a backdoor. The adversarial embedding is directly inserted into the model's embedding matrix, replacing the embedding of a specific, innocuous-looking trigger token (like "!!!!!").

**Questions:**

- The attack forces the model into obvious, low-entropy loops. Have you considered whether a simple output-monitoring defense that detects high token repetition or flags non-progressive reasoning could effectively mitigate this specific manifestation of the attack?

- The backdoor trigger is the token "!!!!!". How sensitive is the attack to the choice of the trigger? For instance, if a more common (but still rare) word was chosen, would the model's pre-existing semantic understanding of that word interfere with the end-to-end training of the adversarial embedding?

- Your objective function $\mathcal{J}_{attack}$ targets transitional tokens after punctuation. Did you experiment with targeting other locations within the reasoning trace or with a different set of "deadlock-inducing" target tokens?

**Ethical Concerns:**

["NO or VERY MINOR ethics concerns only"]

**Final Justification:**

The rebuttal is convincing and i found the paper solid already, so I stand by my final score of 5.

**Limitations:**

The paper's methodology is tailored to exploit the structure of chain-of-thought reasoning, specifically the iterative generation of discrete thought steps. It's unclear how this approach would be adapted to models or tasks that do not follow this explicit "think-then-answer" pattern.


Not really a limitation, but I think you might be interested in these papers:

- Goyal, Sachin, et al. "Think before you speak: Training language models with pause tokens." arXiv preprint arXiv:2310.02226 (2023).
- Kim, Eunki, Sangryul Kim, and James Thorne. "Learning to Insert [PAUSE] Tokens for Better Reasoning." arXiv preprint arXiv:2506.03616 (2025).

**Quality:**

3

**Strengths And Weaknesses:**

Strenghts:

- Comprehensive evaluation across multiple state-of-the-art models, showing a 100% attack success rate across the different LLMs.

- Rigorous investigation of the continuous-to-discrete gap using linear mode connectivity analysis.

- I find very interesting that this methodology exploits the inherent "overthinking" tendency of reasoning models in a sophisticated way.



Weaknesses:

- While the authors claim "minimal impact" on performance for benign inputs, some of the accuracy drops reported in Table 3 are not so small.

- The practical attack relies entirely on the backdoor mechanism, which requires model modification.

---

> ### Author Rebuttal · Authors · 2025-07-31
>
> We are very glad you had a positive initial impression, and we provide point-wise responses below to your questions.
>
> **[Cons 1. Accuracy drops in Table 3?]** Thanks for pointing it out. We offer responses in the following three aspects.
>
> *<Theoretical Stealthiness\>.* Theoretically, our backdoor attack is designed to be stealthy. Our method only modifies the embedding vector for the chosen trigger token (e.g., "!!!!!"). For any benign input that does not contain this specific trigger, the modified embedding parameter is never activated during the model's forward pass. Therefore, the model's behavior should be identical to the original, unattacked model. Our "minimal impact" claim refers to this property and the negligible chance of the trigger appearing accidentally in a benign query.
>
> *<Analysis of Original Results\>.* We attribute the performance fluctuations observed in the original Table 3 to the high variation caused by using a reduced number of test samples. To maintain consistency with our attack evaluation and for computational efficiency, we evaluated stealthiness on the same small test sets (specifically 50 samples each for GSM8K and MMLU-Pro, and the full 43-sample Level 1 subset for MATH500). Such a small sample size can amplify the effect of random variance. As we will demonstrate in the following section, this variance is significantly reduced when a larger number of samples are used for the evaluation.
>
> *<Comprehensive Empirical Re-evaluation>.* To further validate our claim, we conducted a more comprehensive re-evaluation on the Phi-RM and R1-Llama models. We expanded the test sets for the three original benchmarks to 500 samples each and included three additional benchmarks to test generalization: a popular coding benchmark (**HumanEval, Python subset**) and two non-reasoning benchmarks (**MMLU-Pro, Health subset** and **CommonsenseQA**). With this more robust evaluation, the performance differences between the baseline and the backdoored (DA) models are **significantly smaller and fluctuate negligibly** around the baseline across all six benchmarks. This strongly supports our claim that the attack is stealthy and does not systematically degrade model performance. For detailed results, **please see the table in our response to Reviewer Z4Gk's Cons 1**. The choice of two models and a 500-sample size for this new experiment represents a balance between a comprehensive, robust evaluation and the constraints of our available computational resources.
>
> **[Cons 2. Requirement of Model Modification?]** Thanks for pointing out this critical concern. We provide explanations from two perspectives.
>
> *<The Threat Model of Backdoor Attacks\>.* We respectfully clarify that the ultimate goal of our method is to **deliver a poisoned model**, which is consistent with the threat model in most existing backdoor attack work. In this scenario, the attacker acts as the publisher of the model (who could be, but is not necessarily, the original developer), making access to and modification of model weights practical. Many backdoor poisoning attacks [1, 2], which often employ data poisoning, operate under the same assumption. They also modify model weights by fine-tuning the model on a contaminated dataset before its release. As we have demonstrated, our attack is stealthy, meaning the backdoored model behaves identically to the original on benign inputs. The risk arises when a service provider downloads and deploys a released backdoored model, enabling the attacker to launch a resource exhaustion attack at will.
>
> *<Practicality and Plausible Scenarios\>.* A key contribution of our paper is that we identify and experimentally justify the significant **"continuous-to-discrete" adversarial attack gap**, a finding consistent with a recent work [3]. To overcome this challenge, **we propose a backdoor poisoning mechanism that successfully implements the attack**. As illustrated in **Figure 1** of our submission, we provide a practical attack scenario where an attacker implants a backdoor into an open-source model and releases it. When a service provider downloads and uses this model, the attacker can trigger the LLM DoS attack. Furthermore, we can envision other practical scenarios. For instance, a malicious first-party developer could implant a backdoor to **sabotage competitors** building upon their models. Another possibility is to **overcharge users in pay-per-token APIs** by inducing model overthinking to increase the generated token count.
>
> > [1] Xu, Jiashu, et al. "Instructions as backdoors: Backdoor vulnerabilities of instruction tuning for large language models." arXiv preprint arXiv:2305.14710 (2023).
> >
> > [2] Yan, Jun, et al. "Backdooring instruction-tuned large language models with virtual prompt injection." arXiv preprint arXiv:2307.16888 (2023).
> >
> > [3] Rando, Javier, et al. "Adversarial ml problems are getting harder to solve and to evaluate." arXiv preprint arXiv:2502.02260 (2025).
>
> **[Question 1. On Defending via Output Monitoring?]** This is a very good point. We address this in the following two parts.
>
> First, while an output-monitoring strategy could **detect** the attack's effects, implementing it as a practical **defense** or **mitigation** mechanism is non-trivial. Because of the lack of knowledge on the backdoor trigger for the defender, such a defense would need to monitor the model's reasoning process at each step (either per-token or per-segment) for token repetition or non-progressive reasoning and then terminate the generation to prevent the deadlock. However, this per-step detection introduces **significant computational overhead**, which would severely degrade the model's inference efficiency on all benign queries.
>
> On the other hand, while simple token repetition is detectable, rigorously defining and identifying "non-progressive reasoning" is much more difficult. Future work could enhance our attack to be more robust against such defenses, by regularizing the output to be more diverse. A more advanced version could induce a **logical deadlock** in the reasoning process, trapping the model without obvious textual repetition, making the attack far stealthier and harder to mitigate.
>
> **[Question 2. Sensitivity to Trigger Choice?]** Thanks for the insightful question regarding the trigger token.
>
> We claim that our attack is not sensitive to the specific choice of the trigger token. This is because, once the adversarial embedding is optimized (as shown in Fig. 3), it can be projected onto any discrete trigger token through backdoor transplantation.
>
> In other words, our attack intentionally overwrites the original meaning of the chosen token by training the adversarial embedding for it. This semantic alteration is precisely confined to that single token's representation in the model's embedding matrix. For this reason, we recommend using rare tokens (like "!!!!!") as triggers to ensure the modification does not interfere with the model's normal functionality.
>
> **[Question 3. Alternative Attack Targets?]** Thanks for the question. In our preliminary experiments, we did explore a more general optimization, attempting to force various target tokens at every position in the sequence. However, we found this approach ineffective during training where the expressive capacity of a single/few adversarial embedding(s) is limited, failing to consistently induce a deadlock. This led us to develop the more targeted objective presented in the submission，which leverages the inherent structure of the CoT process and select the specific targeting punctuation positions and transitional tokens to make training more effective.
>
> **[Limitation. Generalizability to Non-CoT Models or Tasks?]** Thanks for raising this point. While our method is tailored to exploit the structure of CoT reasoning, we want to clarify that the implementation of our objective function is not strictly limited to a specific model type. Although our initial experiments indicate the attack is most effective on large reasoning models, we want to emphasize that our method is designed to be **input-agnostic**, as the optimization objective targets the general reasoning process itself rather than sample-specific features. This is validated by the fact that the backdoored models show no performance degradation even on non-reasoning benchmarks like MMLU-Pro (Health) and CommonsenseQA, demonstrating the attack's stealthiness. Please see our response to Cons 1 for detailed experimental results and analysis.
>
> **[Related Work]** Thanks for pointing out these two related papers. We will cite them and discuss their connections to and differences from our work in the revised submission.

---

> > ### Comment · Reviewer_u47m · 2025-08-03
> >
> > Thank you for the response!

---

> ### Author Response · Authors · 2025-08-07
> **Thank you!**
>
> Dear Reviewer u47m
>
> Thank you for your diligent review and for checking our response. We greatly appreciate you for keeping the positive rating and for your insightful feedback on our work. We will incorporate the insights from our discussion into the revised version. If you have any more questions, we are happy to discuss them further.
>
> Respectfully,
>
> The Authors

---

### Note · Authors · 2025-08-16

We sincerely thank all reviewers and the Area Chair for their time, valuable feedback, and constructive discussions, which have significantly improved our paper. We are especially grateful for the constructive and in-depth dialogue with Reviewer 5PAW, which has led to an updated rating of our work.


For ease of overseeing the author–reviewer discussion, we summarize below the key points clarified in the rebuttal and discussion:


* Addressed a central concern from Reviewers 5PAW, u47m, and E6ze by justifying the practicality of our white-box threat model, referencing established literature, Trojan AI challenge, and real-world AI supply chain scenarios. This clarification successfully resolved the primary concern of Reviewer 5PAW, who subsequently raised their score from 2.


* Demonstrated the flexibility of our attack in response to detailed feedback from Reviewer 5PAW via **new experiments** with a more realistic, reasoning-centric trigger ("Step-by-step reasoning:"), where we achieved a consistently high attack success rate, addressing their key concern about trigger design.


* Validated the attack's stealthiness in response to questions from Reviewers u47m and 5PAW via **new experiments** through a comprehensive re-evaluation on larger test sets (500 samples) across six diverse benchmarks, where we showed negligible performance impact on benign inputs, resolving concerns about initial performance fluctuations.


* Confirmed the attack's potency beyond model limitations to address questions from Reviewers 5PAW and E6ze via **new experiments** by extending the generation limit to 20k tokens and included a challenging benchmark (AIME 2024) to properly account for false positives, showing our attack significantly outperforms the baseline failure rate.


* Clarified the technical novelty in response to Reviewer E6ze's concerns, highlighting the identification of the "overthinking" vulnerability and the development of a backdoor mechanism to bridge the critical "continuous-to-discrete" adversarial embedding optimization gap, a contribution positively noted by Reviewers u47m and Z4Gk.


Once again, we sincerely thank all reviewers and the AC for their insightful feedback and diligent handling of our submission. Please feel free to reach out if there are any further questions.


Authors

---

### Decision · Program_Chairs · 2025-09-17

**Decision:**

Accept (poster)

**Comment:**

The researchers propose an attack where a model is trained to have backdoor such that, at inference time when the trigger is passed in, the model will reason indefinitely without ever arriving at an answer. As reviewers note, the method is simple and not novel in itself, but it works very well. The paper is also very well-written and easy to understand. The reviewers suggested some additional evaluations, which the authors have completed. They should include these in the next draft of their paper.

Multiple reviewers raised concerns about the threat model not being practical or realistic. The authors addressed this by describing a scenario where they perceive it to be realistic. Like the reviewers, I am not entirely convinced the scenario is especially realistic (it involves an adversary distributing a model, someone else deploying that model, and then the adversary using the attack to cost the deployer wasted compute). However, the fact that this attack works as well as it does is sufficiently interesting that I believe the paper still warrants publication.